# Goals of Care Conversations in Long-Term Care during the First Wave of the COVID-19 Pandemic

**DOI:** 10.3390/jcm11061710

**Published:** 2022-03-19

**Authors:** Laurie Mallery, Nabha Shetty, Paige Moorhouse, Ashley Paige Miller, Maia von Maltzahn, Melissa Buckler, Tanya MacLeod, Samuel A. Stewart, Anne Marie Krueger-Naug

**Affiliations:** 1Department of Medicine, Faculty of Medicine, Dalhousie University, Halifax, NS B3H 4R2, Canada; nabha.shetty@nshealth.ca (N.S.); paige.moorhouse@nshealth.ca (P.M.); ashleyp.miller@nshealth.ca (A.P.M.); maia.maltzahn@nshealth.ca (M.v.M.); annemarie.krueger-naug@nshealth.ca (A.M.K.-N.); 2Nova Scotia Health Authority, Halifax, NS B3H 3A7, Canada; melissa.buckler@nshealth.ca; 3Continuing Professional Development and Medical Education, Halifax, NS B3H 4R2, Canada; tanya@dal.ca; 4Department of Community Health & Epidemiology, Dalhousie University, Halifax, NS B3H 4R2, Canada; sam.stewart@dal.ca

**Keywords:** care planning, COVID-19, frailty, nursing home, prognosis

## Abstract

Goals of care discussions typically focus on decision maker preference and underemphasize prognosis and outcomes related to frailty, resulting in poorly informed decisions. Our objective was to determine whether navigated care planning with nursing home residents or their decision makers changed care plans during the first wave of the COVID-19 pandemic. The MED-LTC virtual consultation service, led by internal medicine specialists, conducted care planning conversations that balanced information-giving/physician guidance with resident autonomy. Consultation included (1) the assessment of co-morbidities, frailty, health trajectory, and capacity; (2) in-depth discussion with decision makers about health status and expected outcomes; and (3) co-development of a care plan. Non-parametric tests and logistic regression determined the significance and factors associated with a change in care plan. Sixty-three residents received virtual consultations to review care goals. Consultation resulted in less aggressive care decisions for 52 residents (83%), while 10 (16%) remained the same. One resident escalated their care plan after a mistaken diagnosis of dementia was corrected. Pre-consultation, 50 residents would have accepted intubation compared to 9 post-consultation. The de-escalation of care plans was associated with dementia, COVID-19 positive status, and advanced frailty. We conclude that during the COVID-19 pandemic, a specialist-led consultation service for frail nursing home residents significantly influenced decisions towards less aggressive care.

## 1. Introduction

During the first wave of the COVID-19 pandemic, residents living in long-term care (LTC) facilities—also known as nursing homes—accounted for more than 80% of COVID-19 deaths in Canada and 97% of deaths in Nova Scotia [1]. The vulnerability of the LTC population highlighted an urgent need to develop care plans that considered the expected effectiveness of interventions and resident preferences. In response, we developed the MED-LTC service, a virtual consultation service led by internal medicine specialists with expertise in acute care medicine, frailty, geriatric medicine, palliative care, and advance care planning. The principal intervention offered by MED-LTC was consultation for care planning to ensure that decisions were appropriate and consistent with well-informed priorities of the decision maker.

### 1.1. Approach to Care Planning

Advance care planning (ACP) and goals of care (GoC) discussions are processes that support adults to consider and communicate their values, life goals, and preferences for future medical care [2]. ACP focuses on future decisions, whereas GoC discussions focus on more imminent situations [3].

Although there is variability in this approach, most care planning guidance suggests placing the individual’s wishes and values at the forefront of the conversation [4]. The typical process includes the following steps:Ask the decision maker to describe their understanding of medical conditions and prognosis;Ask how much information the decision maker wants;Share information on prognosis to the degree desired;Elicit goals, including what is most important;Ask the decision maker to describe their fears and worries about future health;Ask the decision maker about trade-offs, such as what they are willing to accept to possibly gain more time;Develop treatment preferences based on goals.

MED-LTC clinicians identified several challenges with the components of traditional care planning and acknowledged seven barriers, as described below.
*Relaying information to the decision maker.* Many guidelines caution against providing prognostic information [4] and instead, tpically recommend that decision makers describe medical conditions early in the care planning conversation, putting the onus on the individual or their delegate to start the conversation. Yet, many decision makers do not fully appreciate the prognosis associated with frailty, nor the medical conditions that contribute to it [5]. Studies show that receiving realistic information is “very important” to those making medical decisions [6]. Further, providing details about co-morbid conditions and the probability of surviving cardiopulmonary resuscitation (CPR) significantly reduces the number of people who choose resuscitation [7]. Finally, providing information to the decision maker demonstrates the clinician’s understanding of the person’s health story, which allows both parties to consider the full picture. As such, early in the care planning conversation, MED-LTC clinicians described each medical condition, the expected progression of each illness, and the risk/benefit of treatments under consideration.*Making recommendations.* Care planning guidance proposes that physician recommendations “flow only from patient goals” [4]. Yet, when dealing with serious illness, individuals value physician recommendations [6,8]. In support of this evidence, the MED-LTC team made recommendations—especially when treatment had minimal benefit—while still encouraging decision makers to communicate and consider their preferences.*Skill of counselor and time required.* The Respecting Choices^®^ model for care planning advocates a shift from physician counseling to care planning by trained facilitators and community volunteers, such as the clergy [9,10]. This objective does not consider the importance of clinician review of prognostic information, nor the expertise required for care planning conversations. Additionally, care planning recommendations do not address the time and compensation needed for an assessment and discussion related to frailty. MED-LTC assessment and communication typically took several hours.*Values and goals*. Care planning guidance suggests eliciting wishes, values, and preferences early in the conversation (#4–6 above). Instead, MED-LTC clinicians discussed preferences of the decision maker only after they understood the overall clinical prognosis and the risks and benefits of treatment.*Capacity to make medical decisions.* In a study of older patients who required medical decisions, up to 70% did not demonstrate a decision-making capacity [6]. Understanding the decision-making capacity of LTC residents is especially critical, as a significant proportion have dementia [11]. As such, MED-LTC consultations included routine assessment of decisional capacity as part of the care planning process.*Documenting decisions.* Care planning culminates in documented decisions on advance directive forms, such as physician orders for life-sustaining treatment (POLST), which generally do not describe the process used to make decisions. In addition, recorded decisions can be ambiguous, such as whether a ‘Do Not Resuscitate’ (DNR) code status implies a decision about intubation [12]. The lack of detailed, process-oriented documentation presents challenges for the follow-up and implementation of care plans. Thus, MED-LTC consultations included details of the care planning discussion, drivers of the decisions, and comprehensive information about care decisions.*Revisiting decisions.* Care planning typically consists of a single conversation, which result in decisions that remain in effect in perpetuity. The MED-LTC consultation process provided continuous and ongoing support to decision makers to help contextualize decisions when there was a change in health status.

### 1.2. Understanding Evidence

During care planning conversations, MED-LTC consultants delivered information about (1) the expected trajectory and outcomes of baseline conditions, such as frailty, dementia, and co-morbid conditions; and (2) the expected response to SARS-CoV-2 infection and interventions under consideration, such as CPR and intubation. We relied on our collective clinical experience and the following evidence:*Frailty.* Frailty strongly correlates with the risk of morbidity and mortality [13]. It is typically associated with linear, gradual decline [14] in addition to stepwise deterioration following acute illness, after which there may be incomplete recovery and, thus, further frailty [13]. The sequence of decremental decline in health after acute illness can be understood as a frailty cycle (Figure 1) [15].

When establishing care plans, decision makers need to consider this pattern of frailty.
2.*Dementia.* Most residents in nursing homes have dementia [11], a progressive and ultimately fatal condition [16]. To be fully informed, decision makers should understand the diagnosis of dementia, where applicable, and its expected progression.3.*Life expectancy for long-term care residents.* In studies, median survival after admission to a nursing home ranged from 13.7 months to 2.7 years [17], while 1-year mortality ranged between 25% to 35% [18,19]. As such, many LTC residents are nearing the end of life. An important exception to this observation is for individuals with non-progressive disabilities (congenital or acquired).4.*CPR in long-term care facilities.* In 10 studies of LTC residents, survival to hospital discharge after CPR ranged from 0% to 2.9% [20,21,22,23,24,25,26,27]. In three other studies, survival to hospital discharge was between 5% to 13% [27,28]. In a study of individuals with moderate or greater frailty based on the Clinical Frailty Scale, survival to hospital discharge following in-hospital cardiac arrest was 1.8% [29].5.*Prognosis related to SARS-CoV-2 infection during the first wave.* The case fatality rate associated with SARS-CoV-2 infection varies by location and changes over time. Nonetheless, in all situations, mortality increases with age and poor health. During the first pandemic wave, individuals over age 80 with confirmed COVID-19 had a “death rate” of 21.9%, compared to 0.4% for younger individuals [30]. Both frailty [31] and dementia [32] appear to increase mortality.

Non-vaccinated older adults with SARS-CoV2 infection have high mortality rates after intensive care unit (ICU) admission and intubation. At the time of MED-LTC service development, 62% of adults over age 65 years admitted to Seattle hospital ICUs, including both intubated and non-intubated patients, had died [33]. In other studies, mortality after intubation ranged between 68% to 97% [34,35,36,37]. In the only study that showed a comparatively low mortality rate of 20.5% after intubation, the hazard ratio for those 75 years or older was 4.1 (95% (CI, 1.6–10.5; *p* = 0.003) [38].

In a systematic review and meta-analysis [39], COVID-19 patients who had an in-hospital cardiac arrest had a 30-day mortality of 89.9% (95% predicted interval (P.I.) 83.1–94.2%). The estimated overall survival rate with a favorable neurological status at 30 days was 6.3% (95% P.I. 4–9.7%). Following in-hospital cardiac arrest, COVID-19 patients had a higher risk of death compared to those without COVID.

### 1.3. Integrating Prognosis into Care Planning

Based on the above evidence, MED-LTC clinicians concluded that cardiopulmonary resuscitation and intubation with COVID-19 were likely to cause harm with minimal-to-no benefit for frail, older LTC residents. As such, for this population, MED-LTC clinicians communicated these critical facts to decision makers and recommended against CPR and intubation related to COVID-19. We then asked decision makers to consider this perspective when discussing their goals and making decisions. For young residents with non-progressive disabilities, we explained the evidence, engaged in shared decision-making, and provided recommendations where appropriate.

## 2. Materials and Methods

### 2.1. The MED-LTC Team

The MED-LTC clinician team consisted of six internists and one nurse practitioner. All participants had expertise in acute care, geriatric medicine, and palliative care. Before the pandemic, members had established an informal network with a shared interest in frailty-informed care based on the approach established by the Palliative and Therapeutic Harmonization (PATH) program, which aims to help patients and/or families make medical decisions that consider the impact of frailty [40]. In March 2020, when the first wave of COVID-19 disproportionately affected nursing homes, the network was formalized into the MED-LTC team.

### 2.2. Goals of Care Consultation Process

Primary care providers working in Nova Scotia LTC facilities referred residents who were COVID-19 positive (or at risk of becoming positive) and who had either (1) an undocumented care plan or (2) a care plan that represented a mismatch between expected prognosis and selected level of intervention. Consultation included the assessment of medical co-morbidities, frailty stage, cognition, health trajectory, and capacity using a chart review, collateral report (i.e., information from someone who knows the patient), and cognitive testing. If there was uncertainty about the diagnosis or prognosis, the consulting clinician would present the case to MED-LTC team members or other specialists.

Following this comprehensive assessment, there was an in-depth conversation with decision makers about health conditions, expected outcomes related to baseline conditions, and the risks/benefits of potential treatments, followed by the co-development of a care plan. Conversations were conducted using the Nova-Scotia-Health-approved Zoom for Healthcare platform or by phone. Discussion with decision makers followed a semi-structured process and script, adapted from PATH methodology [40] and described in Figure 2 and Figure 3.

### 2.3. Data Sources, Methods, and Variables

Data collection included information about age, gender, frailty level, dementia stage, mobility, function, decisional capacity, and specified level of intervention pre- and post-MED-LTC consultation.

Frailty level was determined using the 9-point, ordinal Clinical Frailty Scale (CFS) [41]. We did not calculate frailty scores for residents under 65 years of age, as the concept of frailty is not as well-validated for the younger population [42]. In particular, mechanisms for functional impairment differ for young individuals, where disability often reflects a single system condition, such as spinal cord injury, or a life-long condition, such as cerebral palsy. In contrast, with advancing age, adults typically develop disability due to the accumulation of health deficits. Thus, the CFS score in a younger person may not confer the same risk as it does for an older person, requiring caution in applying the CFS to younger populations [42].

For statistical analyses, CFS scores were grouped into a binary scale (mild/moderate and severe/very severe) as the score distribution indicated that this was the most statistically appropriate grouping. This methodology has been used elsewhere [43].

Dementia diagnosis was based on a history of progressive functional and cognitive decline, as well as cognitive test results. For statistical analyses, cognitive status was categorized as intact, dementia, or abnormal cognition. The term ‘abnormal cognition’ refers to residents with lifelong non-progressive cognitive disabilities or psychiatric illnesses affecting cognition but without a diagnosis of dementia.

Decisional capacity was based on the resident’s cognitive status and their ability to appreciate, reason, and understand the benefits and risks of proposed treatments. COVID-19 status was based on polymerase chain reaction (PCR) test results.

### 2.4. Goals of Care Outcomes

GoC outcomes were documented using a six-level ordinal scale, as follows:Full code;NO CPR, allow intubation (or did not specify intubation status);NO CPR OR INTUBATION, but allow care in ICU/IMCU;NO CARE IN ICU/IMCU, but allow transfer to hospital;DO NOT HOSPITALIZE; provide full care in LTC;COMFORT CARE ONLY IN LTC.

The first four categories were hospital-based interventions, whereas levels 5 and 6 were delivered in LTC. Patients without documented goals of care (*n* = 3) were assumed to be full code.

### 2.5. Statistical Analysis

Descriptive analysis of patient characteristics included frequencies, means, and standard deviations. A Wilcoxon signed-rank test was used to test for significant changes in GoC levels after consultation, and all models were tested for proportionality using the test of parallel lines. Univariate ordinal logistic regressions were used to determine whether age, frailty stage, cognition, or COVID-19 status were associated with changes in care plans after consultation. All regression models were controlled for pre-intervention GoC level. All analyses were conducted using R version 4.1.0 and IBM SPSS version 26. (R Core Team, Vienna, Austria) and IBM SPSS version 26 (IBM Corporation, Armonk, NY, USA).

The Nova Scotia Health Research Ethics Board approved this research (REB# 26635).

## 3. Results

The service received 64 consults for GoC review from nine LTC facilities from April 2020 to December 2020. Sixty-three consults were completed, as one substitute decision maker declined to participate.

### 3.1. Patient Characteristics

The mean age of participants was 75.3 (SD 14.2). Most consults occurred with residents 65 years of age or older (*n* = 51; 81%). For those 65 years or older, 36 (72%) were severely or very severely frail. Almost all care planning (59/63) was completed with substitute decision makers due to the lack of resident’s decisional capacity or stated preference. Table 1 describes resident characteristics by age group (i.e., those under 65, ≥65, and combined).

### 3.2. Goals of Care Outcomes

After MED-LTC consultation, decision makers chose less aggressive levels of intervention—*p* < 0.001 using a Wilcoxon signed-rank test. The average change in GoC after the intervention was 2.3 levels lower (95% CI: (1.9, 2.7)), with 34 subjects (54%) reducing their GoC by three or more levels.

Fifty-two (83%) chose less aggressive care and ten (16%) remained the same. One resident (1%) escalated their care plan to allow for hospitalization after a mistken diagnosis of dementia was corrected(Table 2, Figure 4).

In the ordinal logistic regression models, dementia, COVID-19 status, and frailty predicted changes in post-intervention GoC (Table 3). Subjects with dementia (or their decision makers) were more likely to choose less aggressive GoC compared to residents without dementia (OR = 4.63). COVID-19-positive status (OR = 4.52) and those with clinical frailty scores of severe or higher (OR = 3.49) were more likely to have less aggressive GoC. The test of parallel lines found no evidence of violations of the proportionality assumption (*p*-values ranged from 0.29–0.99).

We performed a sensitivity analysis with the population restricted to ages 65+ (data not shown) and found no noticeable difference in modeling results.

An analysis of interviews from long-term care staff provided overwhelmingly positive feedback about the MED-LTC intervention and supported the need for a consultation service to address goals of care. A separate publication will report on the qualitative experience of the program from a clinician perspective.

## 4. Discussion

The COVID-19 pandemic presented an opportunity to develop a virtual, specialist consultation service in LTC facilities. Our findings demonstrate that during this time, MED-LTC consultation was associated with a significant change in care decisions towards less aggressive interventions.

Compared to standard guidance on how to conduct care planning discussions, MED-LTC consultations began with a comprehensive assessment of the resident’s co-morbidities and frailty stage. We then provided detailed information to the decision maker about medical conditions, anticipated health trajectory, and expected outcomes of interventions under consideration. When a resident was severely frail and/or COVID-19 positive, based on published data, the team regularly made recommendations against CPR, in general, and intubation with COVID-19. When outcomes were less clear, such as for younger residents or those with non-progressive disabilities, we provided navigation and did not always make concrete recommendations. Consultants then assisted decision makers in applying the values and goals of the resident to the medical information and advice, culminating in individualized, appropriate care planning. With this approach, 84% of decision makers chose less aggressive interventions, influenced by dementia, COVID-19-positive status, and more advanced frailty. Although the catalyst for the MED-LTC service was the COVID-19 pandemic, the process is also relevant for medical decision making unrelated to COVID-19.

The results of this study are in line with previous studies. In an evaluation of the PATH program, which formed the basis for the MED-LTC approach to decision-making, 75% of frail patients (or their decision makers) decided not to pursue proposed surgery [40]. Likewise, in a study of 18 SARS-CoV-2-positive nursing home residents, structured communication changed decisions [44]. After discussion, 9 (52%) residents chose a do-not-hospitalize status compared to 1 (6%) at baseline. DNR designations increased from 7 (41%) to 15 (88%). Similar to the MED-LTC process, care planning discussions relayed health information early in the conversation. However, unlike our approach, clinicians did not make recommendations and instead posed questions such as, “Would you want us to perform chest compressions on your loved one?”. This type of question is in accord with care planning guidance, which does not typically endorse clinician recommendations, even when a treatment has little-to-no chance of success. In contrast, MED-LTC clinicians aimed to balance concerns about paternalism with the burden that substitute decision makers might feel in making decisions without guidance [45].

Our study has several limitations. First, the sample size of 63 residents is small. The subgroup analysis (Table 3) must be cautiously interpreted, as the sample size in each group is even smaller. This study is limited to a quantitative evaluation of the care planning intervention. A separate publication will report on the qualitative experience of the program from a clinician and patient/decision maker perspective. The study included a sample of residents/families in long-term care from Eastern Canada, all of whom had disabilities and co-morbidities, which might limit generalizability to other populations. The approach described in this paper is best suited to populations with similar medical complexity.

The MED-LTC care planning process is time- and resource-intensive. As such, the approach may not be feasible for practitioners who work in a fee-for-service reimbursement model. Yet, this study shows that, when clinicians invest time and effort in advance care planning, it can significantly limit non-beneficial care. Care plans that align with frailty and prognosis promote system cost-effectiveness by reducing expensive high-intensity intervention at the end of life [46].

## 5. Conclusions

During the COVID-19 pandemic, a specialist-led consultation service in long-term care to discuss care goals with residents or their substitute decision makers significantly influenced decisions towards less aggressive care. Based on the evidence presented in this study, we conclude that care planning should include an in-depth clinician review of medical conditions; consideration of capacity; full disclosure of prognosis to decision makers with recommendations when applicable; and improved documentation of decisions. These steps may be missing in many goals of care discussions, particularly the provision of comprehensive, realistic medical information to facilitate informed consent. As care planning gains more attention, this study shows that the process needs to be executed carefully, with proper skill, expertise, and an appropriate timeframe especially for populations with medical complexity.

## Figures and Tables

**Figure 1 jcm-11-01710-f001:**
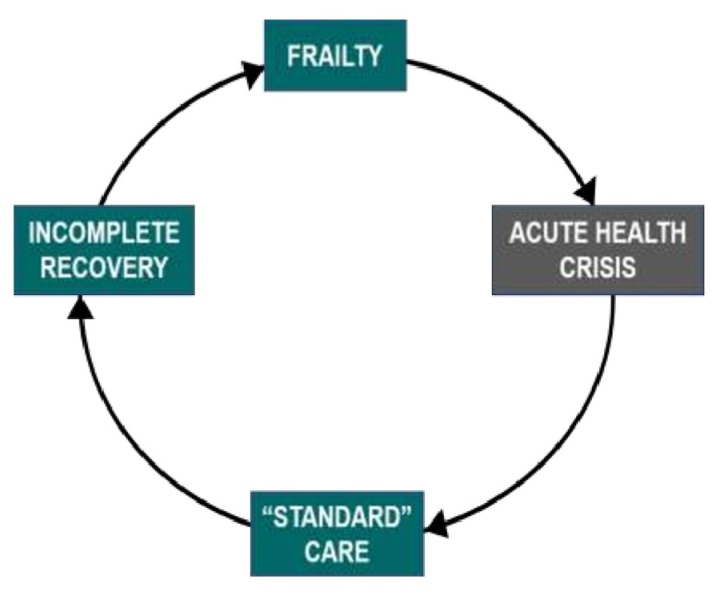
The frailty cycle.

**Figure 2 jcm-11-01710-f002:**
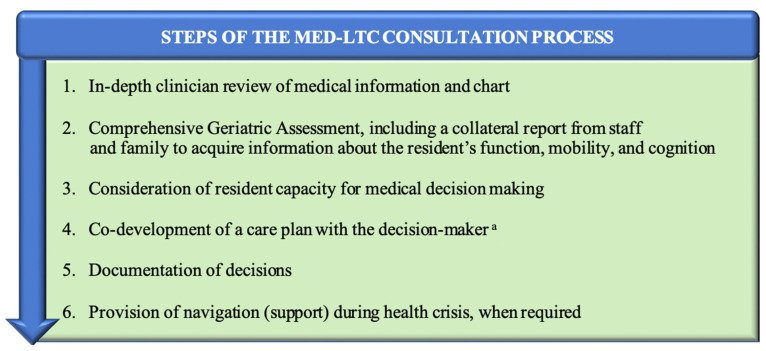
Steps of the Med-LTC Consultation process. ^a^ See Figure 3.

**Figure 3 jcm-11-01710-f003:**
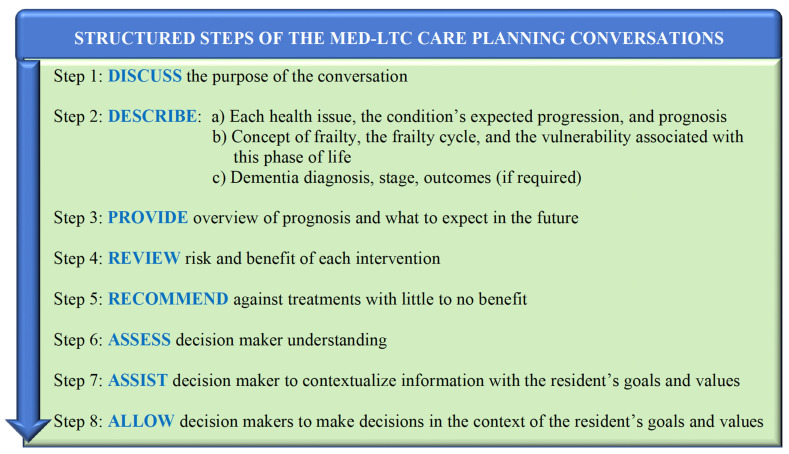
Structured steps of the Med-LTC care planning conversations.

**Figure 4 jcm-11-01710-f004:**
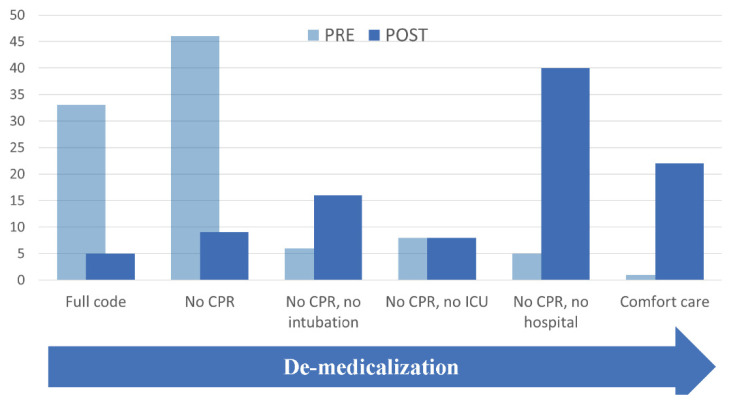
Decisions pre- vs. post-MED-LTC consultation (*n* = 63).

**Table 1 jcm-11-01710-t001:** Characteristics of long-term care residents.

Characteristic		Age Groups (years)		
		All Combined (*n* = 63)	Under 65 (*n* = 12)	65 and Over (*n* = 51)
Positive for COVID-19 at Consult—No. (%)		28 (44.4%)	6 (50%)	22 (43)
Age—year	Mean (SD)	75.3 (14.2)	51.8 (7.8)	80.9 (8)
	Median (range)	77.0 (37–96)	52.0 (37, 64)	81.0 (65, 96)
Female Sex—No. (%)		41 (65)	8 (66.7%)	33 (65)
Clinical Frailty Score	Mild/moderate	N/A ^a^	N/A ^a^	14 (28)
	Severe/very severe	N/A ^a^	N/A ^a^	36 (72)
Cognitive status—No. (%)	Dementia	36 (57)	2 (17)	34 (67)
	Abnormal cognition ^b^	16 (25)	9 (75)	7 (14)
	Intact cognition	11 (18)	1 (8)	10 (20)
Mobility—No. (%)	No aid	8 (13)	2 (17)	6 (12)
	Gait aid/needs assistance	21 (34)	2 (17)	19 (38)
	Cannot walk	33 (53)	8 (67)	25 (50)
Basic Activities of Daily Living —No. (%)	Independent	3 (5)	0 (0)	3 (6)
	Dependent for 1–2 activities	21 (33)	2 (16)	19 (37)
	Dependent for 3 or more activities	39 (61)	10 (83)	29 (57)

^a^ N/A = frailty score is not recorded for the younger population due to a lack of validation of the Clinical Frailty Scale for those under 65 years of age (see above discussion). ^b^ Abnormal cognition describes residents with lifelong, non-progressive cognitive disabilities or psychiatric illness affecting cognition but without a diagnosis of dementia.

**Table 2 jcm-11-01710-t002:** Goals of care outcomes by level (*n* = 63).

LEVEL	Pre-Consult Level No. (%)	Post-Consult Level No. (%)
1. Full code	24 (33.4) ^a^	3 (4.8)
2. NO CPR, allow intubation (or did not specify intubation status)	29 (46.0)	6 (9.5)
3. NO CPR OR INTUBATION, but allow care in ICU/IMCU	4 (6)	10 (16)
4. NO CARE IN ICU/IMCU, but allow transfer to hospital	5 (8)	5 (8)
5. DO NOT HOSPITALIZE; provide full care in LTC	3 (5)	25 (40)
6. COMFORT CARE ONLY IN LTC	1 (2)	14 (22)

^a^ Charts without documented care goals (*n* = 3) were assumed to be full code. CPR = cardiopulmonary resuscitation. ICU = intensive care unit. IMCU = intermediate care unit. LTC = long-term care.

**Table 3 jcm-11-01710-t003:** Factors predicting decisions for less aggressive care post-intervention.

Variable		OR (95% CI)
Age group, (years)	<65	1, ref.
	65 and older	1.82 (0.55, 6.02)
Cognitive status	No dementia	1, ref.
	Dementia	4.63 (1.03, 20.92)
	Abnormal cognition ^a^	0.80 (0.17, 2.64)
Frailty stage according to CFS	Moderate or lower	1, ref.
	Severe or higher	3.49 (1.01, 12.09)
Positive for COVID-19	No	1, ref.
	Yes	4.52 (1.64, 12.42)

CI = confidence interval. ref. = reference. ^a^ Abnormal cognition describes residents with lifelong non-progressive cognitive disabilities or psychiatric illness affecting cognition but without a diagnosis of dementia. CFS = Clinical Frailty Scale.

## Data Availability

Datasets are available on request.

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
