# Peer review of "Goals of Care Conversations in Long-Term Care during the First Wave of the COVID-19 Pandemic"

_jcm, 2022, doi:10.3390/jcm11061710_

Round 1
Reviewer 1 Report
This paper aims to investigate whether old LTC residents (or their decision-makers) in nursing-homes would change their care plans after being informed with key facts about CPR and intubation by MED-LTC team during the first wave of the Covid-19 pandemic. 64 patients were included in the study and information including age, gender, frailty level, dementia stage,mobility, function, decisional capacity, and specified level of intervention pre- and post- MED-LTC consultation were collected. Statistical methods (Wilcoxon Signed Rank Test and Ordinal Logistic Regressions) were used to investigate the patients’ decision change in GoC levels before and after consultation, as well those factors predicting patients’ decision. Overall, the topic of this study is quite relevant and significant during the covid-19 pandemic.
There are some suggestions which would be helpful to revise the paper:
- P4, In section 1.3, the authors claimed that “The MED-LTC communicated these key facts to decision-makers and recommended against cardiopulmonary resuscitation and intubation related to Covid-19”, did the consultant told different patient exactly the same information? Or case by case (considering each patient’s specific situation)? The author should provide more detailed information about the physician guidance, or give samples to help us understand the process.
- In Table 1, 22(43) in Column “Positive for Covid-19 atconsult—No. (%)” should be 22(43%); I suggest to keep one decimal place after the decimal point for % numbers. Clinical Frailty Score was not recorded for patients under 65,why? Also please check format and numbers in Columns “Female sex—No. (%)”,”Cognitive status—No. (%)” ,”Cognitive status—No. (%)”,”Basic Activities of Daily Living—No. (%).
- The paper reported the patients’GoC levels and claimed that “The average change in GoC after the intervention was 2.3 levels lower, with 34 subjects (54%) reducing their GoC by three or more levels“. This indicates that most patients chosen less aggressive care decisions, but we still know little about the health well-being for those patients, like patients’ satisfaction, death rate, expense and so on. Is is possible to display these kind of information?
- Before the Ordinal Logistic Regressions, the authors need to Test of parallel lines. Meanwhile, Collinearity diagnosticsbetween Predicting Factors is recommended to conduct so as to test whether there is multicollinearity between variables.

Reviewer 2 Report
- The present study is a good analysis of care plans and conservation during COVID-19, however, authors need to address the following questions in more straightforward details.
- What will be the outcome, relevance, and future implications of the study?
- Also, this study is limited to the specific set of patients from a specific region, then what is the implication of this study for others? Please justify.
- The conclusion section is just a few lines. The authors need to detail the conclusion a bit.
- Authors need to provide a few statistical analysis-based pie charts or figures that will make this study more clear understandable to readers.
Author Response
See the attached attachment

Round 2
Reviewer 2 Report
The authors successfully replied to all the reviewer's comments.